# Efficient Multilingual Language Model Compression through Vocabulary Trimming

**Asahi Ushio** and **Yi Zhou** and **Jose Camacho-Collados**
Cardiff NLP, School of Computer Science and Informatics, Cardiff University, UK
{UshioA,Zhouy131,CamachoColladosJ}@cardiff.ac.uk

## Abstract

Multilingual language models (LMs) have become a powerful tool in NLP, especially for non-English languages. Nevertheless, model parameters of multilingual LMs remain large due to the larger embedding matrix of the vocabulary covering tokens in different languages. Instead, monolingual LMs can be trained in a target language with the language-specific vocabulary only. In this paper, we propose *vocabulary-trimming* (VT), a method to reduce a multilingual LM vocabulary to a target language by deleting potentially irrelevant tokens from its vocabulary. In theory, VT can compress any existing multilingual LM to any language covered by the original model. In our experiments, we show that VT can retain the original performance of the multilingual LM, while being considerably smaller in size than the original multilingual LM. The evaluation is performed over four NLP tasks (two generative and two classification tasks) among four widely used multilingual LMs in seven languages. The results show that this methodology can keep the best of both monolingual and multilingual worlds by keeping a small size as monolingual models without the need for specifically retraining them, and can even help limit potentially harmful social biases.

## 1 Introduction

Multilingual language model (LM) pre-training (Devlin et al., 2019; Conneau et al., 2019; Liu et al., 2020; Xue et al., 2021) has been shown to be an efficient mechanism to store information from many languages into a single model, without the need for training multiple language-specific models. Moreover, it has been proven reliable for cross-lingual tasks (Pires et al., 2019; Conneau and Lample, 2019) and can provide competitive performance in most settings, generally similar to its monolingual counterparts (Goyal et al., 2021), while being generally less affected by culturally-dependant biases (Ahn and Oh, 2021). Similarly to monolingual

models, multilingual LMs can be used for zero/few-shot learning (Scao et al., 2022) by increasing the model size and, more frequently, can be specialized to different tasks by fine-tuning to specific data. In practice, however, there are a few practical issues when training multilingual LM such as the curse of multilinguality (Conneau et al., 2019; Pfeiffer et al., 2022), a trade-off between the number of languages and individual performance in a single language, or the multilingual vocabulary construction, which requires a careful design for better generalization (Chung et al., 2020; Zheng et al., 2021; Liang et al., 2023).

Besides such generalization concerns, multilingual LMs usually consist of larger parameters than their monolingual counterparts due to the need for a large vocabulary covering multiple languages. This becomes an important issue in practice when the resources to host models are limited. For instance, while using the same configuration (i.e., same number of layers and hidden units), the parameter size of T5$_{\text{SMALL}}$ (Raffel et al., 2020) and mT5$_{\text{SMALL}}$ (Xue et al., 2021) are 140M and 300M, respectively. This is only due to their difference in vocabulary size, with T5 being 50k and mT5, 250k. In fact, the embedding matrix stemming from the LM vocabulary can occupy a large portion of the parameter space. For instance, the ratio of the embedding matrix to the full model's parameter size in multilingual LMs can be higher than 80% as T5 (see Figure 1).

In this paper, we propose a simple *vocabulary trimming (VT)* method to remove tokens from the vocabulary of multilingual LMs that may be irrelevant to the target language.[1] This is achieved by automatically identifying language-specific tokens from an underlying text corpus. We consider two VT strategies of pre-FT VT (VT before fine-tuning) and post-FT VT (VT after fine-tuning) and

---

[1]We release a Python library for VT at `https://github.com/asahi417/lm-vocab-trimmer`.

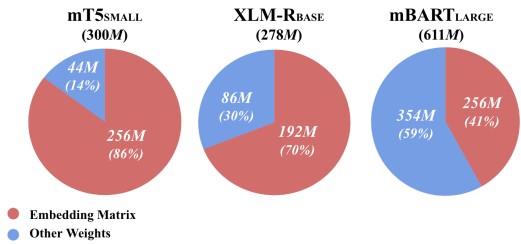

Figure 1: The ratio of the embedding matrix to the number of parameters for each multilingual LM.

analyse them by varying the final vocabulary size. We conduct experiments on two generation tasks, question answering (QA) and question generation (QG), and two classification tasks, sentiment analysis and natural language inference (NLI), across seven different languages. The experimental results show that both pre and post fine-tuning VT can reduce the model size while retaining the original performance in generation tasks (QA and QG), and particularly in classification tasks (sentiment and NLI) where the results are close to being identical despite the significant reduction in vocabulary size. In all tasks, the original performance can be generally maintained with less than 40% of the full model parameters for all languages.

Finally, even though pre-trained LMs have reported impressive performance on various NLP downstream tasks (Kenton and Toutanova, 2019; Liu et al., 2019; Conneau et al., 2019), such LMs also demonstrate worrying levels of social biases in certain situations (May et al., 2019; Kurita et al., 2019; Kaneko and Bollegala, 2021). One natural question that arises is whether VT can have an influence on the bias level in multilingual LMs, including fine-tuned models. For this purpose, we evaluate social bias in multilingual LMs after applying VT with different settings and compare it against its monolingual counterpart. Experimental results show that the monolingual LM tends to contain more bias than its multilingual versions. Moreover, compared to the original multilingual LM, the bias level has no significant change after applying VT. These results suggest that a monolingual LM can be induced by applying VT to its corresponding multilingual LM, thereby obtaining a less biased monolingual LM compared to its original monolingual counterpart.

## 2 Related Work

Several studies have explored the possibility to modify or adapt the vocabulary of LMs. For instance, Artetxe et al. (2020) and Marchisio et al. (2022) adapted a mono-lingual LM into another language by learning the embedding matrix on the new language, while fixing the other weights. Similarly, Wang et al. (2019) augmented the vocabulary of a multilingual LM to new languages with multilingual word alignment (Lample et al., 2018). Zheng et al. (2021) proposed to evaluate the ability of a vocabulary to represent a particular language, and Chung et al. (2020) proposed a multilingual vocabulary construction that balances the trade-off between optimizing for cross-lingual sub-word sharing and the need for robust representation of individual languages. XLM-V (Liang et al., 2023) combines the idea of Zheng et al. (2021) and Chung et al. (2020) to efficiently enlarge the vocabulary size along with the model size scaling. Ostendorff and Rehm (2023) used a multi-stage fine-tuning to obtain a LM in the target language from other LM in the source language. These prior works modify existing mono/multi-lingual LMs to include new languages, i.e. augmenting the multilinguality of the LMs. In contrast, our study focuses on compressing multilingual LMs into the target language to effectively achieve smaller monolingual LMs, i.e. reducing the multilingual representation of the LMs while retaining the capability in a specific target language.

The work of Abdaoui et al. (2020) is the most relevant to our study as, to the best of our knowledge, they introduced the idea of VT for the first time. However, their analysis is limited to NLI with pre-fine-tuning VT with mBERT (Devlin et al., 2019) only, as well as a fixed vocabulary size after VT. In contrast, our study compares two VT strategies, before and after fine-tuning, and show how this latter strategy, not considered in Abdaoui et al. (2020), can be a more effective compression technique in some settings. Furthermore, we extend the experiments to generation tasks as well as classification tasks with more recent LMs such as mBART (Lewis et al., 2020) and mT5 (Xue et al., 2021), and provide an exhaustive analysis on the effect of VT.

## 3 Vocabulary Trimming

To perform vocabulary trimming (VT), we first need a multilingual LM as an input. The idea is to

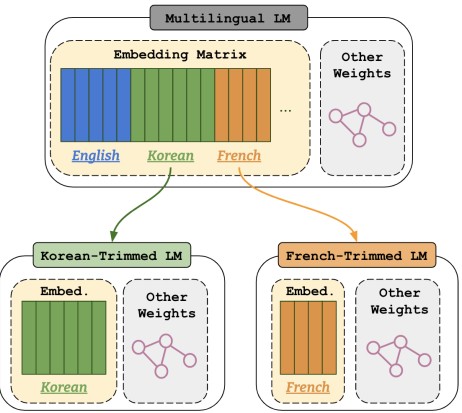

Figure 2: An illustration of vocabulary trimming for Korean and French.

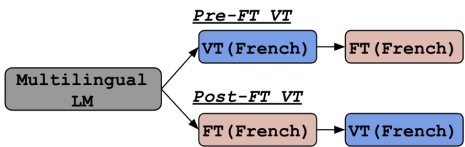

Figure 3: Comparisons of *Pre-FT* vs *Post-FT* VT in an example of fine-tuning on a task in French.

tailor model to a particular target language $l$, which in principle belong to the same set of languages used to trained the input multilingual LM.[2] For the target language $l$, VT first identifies language-specific tokens on a language-specific corpus $\mathcal{C}_l$, and remove all the tokens along with their embeddings except for those appeared in $\mathcal{C}_l$ as described in Figure 2. In our analysis (§ 5), we also consider to keep the top-$n$ most frequent tokens in $\mathcal{C}_l$ to further reduce the model size by removing less frequent tokens. We consider two VT strategies: (1) before fine-tuning and (2) after fine-tuning.

The difference between these two strategies is whether to perform VT before or after fine-tuning, as shown in Figure 3. Both VTs have advantages and drawbacks: while pre-FT VT can reduce the time of fine-tuning as the trimmed LM is smaller than the original LM, post-FT VT only need a fine-tuned multilingual LM - this way, post-FT VT can be used as a postprocessing step and no additional language-specific training is required.

Finally, we release a simple LM vocabulary trimming starting package to apply our proposed technique to any input multilingual transformer-based LM, along with all the models and code needed to reproduce our experiments, at `https://github.com/asahi417/lm-vocab-trimmer`.

## 4 Evaluation

In this section, we present our experimental results to test the reliability of our proposed VT methodology in NLP tasks.

---

[2]In theory, vocabulary trimming could be applied to any language model, even monolingual, but this analysis is out of the scope of this paper.

### 4.1 Experimental Setting

**Tasks and datasets.** In order to test the efficacy of VT, we consider two generation tasks, question answering (QA) and question generation (QG), and two classification tasks, sentiment analysis and natural language inference (NLI). As the datasets for QA, we use SQuAD (Rajpurkar et al., 2016) (English), Spanish SQuAD (Casimiro Pio et al., 2019) (Spanish), FQuAD (d'Hoffschmidt et al., 2020) (French), Italian SQuAD (Croce et al., 2018) (Italian), JAQuAD (So et al., 2022) (Japanese), Ko-rQuAD (Lim et al., 2019) (Korean), and SberQuAd (Efimov et al., 2020) (Russian). For QG, we use the same datasets adapted for QG via QG-Bench (Ushio et al., 2022). For sentiment analysis, we use Twitter-based datasets for English (Rosenthal et al., 2017), Arabic (Rosenthal et al., 2017), French (Benamara et al., 2017), Italian (Barbieri et al., 2016), German (Cieliebak et al., 2017), Portuguese (Brum and Nunes, 2017), and Spanish (Díaz-Galiano et al., 2018) from UMSAB (Unified Multilingual Sentiment Analysis Benchmark) (Barbieri et al., 2022). All the sentiment analysis datasets contain three labels: positive, neutral and negative. For NLI, we use XNLI (Conneau et al., 2018), a multilingual NLI dataset, including English, French, German, Spanish and Arabic, which are the languages included in the sentiment analysis experiment. We fine-tune LMs on the training sets of each language, which were translated automatically from English and released in the original paper.

**Evaluation metrics.** For the evaluation, we use the following standard metrics: answer span F1 score (Ans-F1) and exact match (EM) are used for QA; METEOR (MTR) and BERTScore (BS) for QG, which have been shown to be the most correlated metrics to human judgment (Ushio et al., 2022); macro-F1 score for sentiment following (Barbieri et al., 2022); and accuracy for NLI. As the language-specific corpus $\mathcal{C}_l$ to extract vocabulary counts for VT, we use mC4 (Xue et al., 2021), one of the largest public multilingual corpora.

| | Lang. | Vocabulary | Parameters | No-Trim | QA Post-FT | Pre-FT | No-Trim | QG Post-FT | Pre-FT |
|---|---|---|---|---|---|---|---|---|---|
| mT5 | EN | 209K (83.6%) | 258M (86.1%) | 70.1 / 55.5 | **70.2** / 55.5 | 70.1 / **56.4** | 23.8 / 90.0 | 23.8 / 90.0 | **24.0 / 90.1** |
| | ES | 131K (52.4%) | 178M (59.4%) | 55.9 / 34.7 | 55.9 / 34.7 | **57.8 / 37.5** | 22.7 / 84.1 | **22.7** / 84.1 | 22.3 / **84.2** |
| | FR | 131K (52.4%) | 178M (59.4%) | **50.0 / 30.9** | **50.0 / 30.9** | 48.6 / 29.4 | 17.5 / 80.7 | **17.5 / 80.7** | 16.1 / 79.2 |
| | IT | 111K (44.4%) | 157M (52.6%) | 53.2 / 37.6 | **53.4 / 37.8** | 51.5 / 36.0 | 17.6 / 80.8 | **17.6 / 80.8** | 17.5 / 80.6 |
| | JA | 125K (50.0%) | 172M (57.6%) | **65.7 / 65.7** | **65.7 / 65.7** | 63.0 / 63.0 | 29.0 / 80.9 | **29.0** / 80.9 | 28.6 / **81.0** |
| | KO | 73K (29.2%) | 119M (39.7%) | **77.1 / 70.6** | **77.1** / 70.5 | 74.5 / 67.3 | 27.5 / 82.9 | 27.5 / 83.0 | **28.0 / 83.7** |
| | RU | 147K (58.8%) | 195M (65.1%) | 73.7 / 51.4 | 73.8 / 51.4 | **74.8 / 53.4** | 26.4 / 84.3 | 26.4 / 84.3 | **28.9 / 86.4** |
| mBART | EN | 173K (69.2%) | 532M (87.1%) | 76.9 / 62.6 | 77.0 / 62.7 | **78.4 / 65.7** | **25.1 / 90.4** | **25.1 / 90.4** | 24.7 / 90.1 |
| | ES | 87K (34.8%) | 443M (72.7%) | 64.1 / 42.2 | **64.5** / 42.8 | 63.7 / **43.9** | **22.9** / 83.6 | 22.8 / 83.6 | 22.8 / **84.0** |
| | FR | 85K (34.0%) | 442M (72.5%) | 60.4 / 39.3 | 61.0 / 39.8 | **66.4 / 45.1** | 19.8 / 81.7 | **19.8 / 81.7** | 18.4 / 79.7 |
| | IT | 67K (26.8%) | 424M (69.5%) | 64.7 / 50.0 | 64.9 / **50.2** | **65.8** / 49.8 | 18.0 / 80.6 | 17.9 / 80.7 | **18.9 / 81.1** |
| | JA | 77K (30.8%) | 434M (71.1%) | 68.2 / 68.2 | 68.2 / 68.2 | **70.6 / 70.6** | **30.0** / 82.3 | 29.7 / 82.1 | 29.1 / 80.8 |
| | KO | 46K (18.4%) | 402M (65.9%) | 79.3 / 72.3 | 79.2 / 72.1 | **83.2 / 77.3** | 30.2 / 83.9 | **30.3 / 84.0** | 30.2 / 83.8 |
| | RU | 99K (39.6%) | 456M (74.8%) | 78.7 / 58.0 | **79.0 / 58.2** | 75.5 / 49.9 | **29.3 / 87.2** | 28.7 / 87.0 | 28.3 / 86.7 |

Table 1: Results on QA (Ans-F1/EM) and QG (MTR/BS), including both the vocabulary size and the number of parameters after VT with the ratio to the original model (%). The best results in each LM and language are in bold characters. Note that the parameter size of the original mT5 and mBART (No-Trim) is 300M and 611M, respectively, both with a vocabulary size of 250K.

**Base language models.** As the base LMs, given computational constraints we chose the smallest mT5 and mBART to fine-tune on QA and QG, and XLM-R and XLM-V (Liang et al., 2023) for sentiment analysis and NLI. All these models have a vocabulary size of 250K, except for XLM-V which has a vocabulary size of 901K subword tokens. For our experiments, we compare the results of pre/post-FT VT against vanilla LM fine-tuning without VT, which we refer to as *No-Trim*.

**Fine-tuning.** For model fine-tuning, we rely on lmqg (Ushio et al., 2023) for QA/QG, and Ray Tune[3] for sentiment analysis. In both cases, we use the default search space for hyperparameter search. For NLI, we follow the same hyperparameters used in Liang et al. (2023). All the resulting models and code will be released upon acceptance of the paper.

## 4.2 Results

We present the results for the generation and classification tasks in section 4.2.1 and section 4.2.2, respectively.

### 4.2.1 Generation Tasks: QA & QG

Table 1 shows the overall results on QA and QG. The results confirm that both of pre/post-FT VT can maintain the original performance in most cases, while being smaller than the original models by significantly reducing the vocabulary size. First, post-FT VT achieves at least the same performance as

---

[3] https://docs.ray.io/en/latest/tune/index.html

the vanilla fine-tuning for all the languages for both LMs in QA and QG, except for a few cases such as mBART QA in Korean and mBART QG in Russian, although the decrease is no more than 0.5%. Meanwhile, pre-FT VT outperforms its vanilla fine-tuning model with a relatively important margin in some cases, such as mBART French QA and mT5 Spanish QA. In contrast, there are a few models where pre-FT VT degrades the performance of the original model such as mT5 QA in Korean (2.6% decrease in Ans-F1) or mBART QA in Russian (3.2% decrease in Ans-F1).

Since we keep all the vocabulary that appeared in the language-specific corpus $\mathcal{C}_l$, the percentage of reduced parameter depends on the language, and generally VT can reduce the model size for Asian (Japanese/Korean) and European (Spanish/French/Italian) languages efficiently (50% for mT5 and 70% for mBART), but it remains high in other languages (English/Russian).

### 4.2.2 Classification Tasks: Sentiment & NLI

Table 2 shows the results on sentiment analysis and NLI. In this case, post-FT VT can robustly preserve the original performance of the original No-Trim baseline in both tasks for XLM-R and XLM-V, while being no more than 40% and 60% in vocabulary and overall parameter size, respectively, of the original XLM-V and XLM-R models in all the non-English datasets. XLM-V PT sentiment model is the only post-FT VT where a slight decrease can be observed (0.1%). On the other hand, the accuracy

| | Lang. | Vocabulary | Parameter | Sentiment | | | NLI | | |
|---|---|---|---|---|---|---|---|---|---|
| | | | | No-Trim | Post-FT | Pre-FT | No-Trim | Post-FT | Pre-FT |
| XLM-R | AR | 49K (19.6%) | 124M (44.7%) | **66.3** | **66.3** | 60.9 | **75.7** | **75.7** | 73.8 |
| | DE | 91K (36.4%) | 156M (56.3%) | 73.2 | 73.3 | **73.5** | **79.9** | **79.9** | 78.3 |
| | EN | 173K (69.2%) | 219M (78.7%) | 68.4 | **68.5** | 67.9 | **84.6** | **84.6** | 70.6 |
| | ES | 87K (34.8%) | 153M (55.0%) | **69.0** | **69.0** | 65.0 | **79.8** | **79.8** | 67.2 |
| | FR | 85K (34.0%) | 151M (54.6%) | 71.8 | 71.8 | **72.1** | **80.1** | **80.1** | 79.6 |
| | IT | 67K (26.8%) | 138M (49.7%) | 62.9 | 62.9 | **70.8** | - | - | - |
| | PT | 66K (26.4%) | 137M (49.3%) | 70.7 | **70.8** | 70.2 | - | - | - |
| XLM-V | AR | 92K (11.8%) | 157M (20.2%) | 59.8 | 59.8 | **64.7** | 75.5 | 75.6 | **76.1** |
| | DE | 239K (30.7%) | 269M (34.7%) | **73.5** | **73.5** | 73.0 | 78.9 | 78.9 | **79.0** |
| | EN | 484K (62.2%) | 458M (58.9%) | **63.9** | **63.9** | 61.3 | 84.4 | 84.4 | **84.5** |
| | ES | 243K (31.2%) | 279M (35.1%) | 60.7 | 60.7 | **66.6** | **80.7** | **80.7** | 80.6 |
| | FR | 218K (28.0%) | 253M (32.6%) | **68.8** | **68.8** | 59.5 | 78.6 | 78.6 | **79.0** |
| | IT | 184K (23.7%) | 227M (29.3%) | 70.2 | 70.2 | **74.2** | - | - | - |
| | PT | 181K (23.3%) | 225M (28.9%) | **66.6** | 66.5 | 52.8 | - | - | - |

Table 2: Results of sentiment analysis (macro F1) and XNLI (accuracy) including both the vocabulary size and the number of parameters after VT with the ratio to the original model (%). The best results in each LM and language are in bold characters. Note that the overall parameter size of the original XLM-R and XLM-V (No-Trim) is 278M and 778M, respectively, with the vocabulary size being 250K and 901K vocabulary in each case.

of Pre-FT VT appears to be sensitive to the language and task, where it improves the performance in some languages such as Italian (XLM-R and XLM-V achieve 7.9% and 3.8% increase for sentiment analysis), but it decreases the performance with non-trivial margin in other languages such as Arabic, where XLM-R decreases 5% for sentiment analysis and 2% for XNLI. Since XLM-V has a larger vocabulary size, the percentage of reduced parameters at VT is more prominent in XLM-V, as seen in Arabic (20.2%) and Portuguese (28.9%) for example.

## 5 Vocabulary Size Analysis

In our main experimental results (§ 4.2), all the unique tokens that appeared in the monolingual corpus were kept, which resulted in a low compression ratio for some languages such as English and Russian. In this analysis, we constrain the number of vocabulary and choose the top-$n$ vocabulary at VT in terms of the frequency in the corpus (see § 3). For QA and QG, we compare mT5$_{SMALL}$ results with $n$ from [5K, 10K, 15K, 30K, 60K, 90K, 120K], which correspond to an overall parameter size of [49M, 54M, 59M, 74M, 105M, 136M, 166M], respectively. For sentiment analysis and NLI, we experiment with XLM-R$_{BASE}$ with $n$ from [5K, 10K, 15K, 30K, 60K], which correspond to [89M, 93M, 97M, 109M, 132M] of parameter size, respectively[4].

[4]The full results with top-$n$ can be found at Appendix B and Appendix A.

### 5.1 Generation Tasks: QA & QG

Figure 4 shows the results of mT5 on QA and QG. Noticeably, post-FT VT can reduce the vocabulary size to 60K for both QA and QG in all the languages with a trivial gap (0.3% decrease of EM in Russian QA and 0.1% decrease of BS in French QG), and that is **35%** of the original mT5 in the parameter size. Furthermore, post-FT VT can further reduce the vocabulary to 5K tokens with no more than 0.4% decrease in each metric for both QA and QG in English, French, and Italian, which is **16%** of the original mT5 in the parameter size. Meanwhile, pre-FT VT outperforms the No-Trim result in all the languages in QA, and the majority of the languages in QG (English, Italian, Korean, and Russian), but the result is sensitive to the choice of $n$. For example, Japanese/Korean QA and Russian QG with pre-FT VT for top-5K (16% of the original mT5) outperforms No-Trim as well as post-FT VT, but Japanese QG with pre-FT VT is worse in any choice of $n$ on contrary. This larger variation of results may also be due to the parameter size space, as the optimal parameters for the original multilingual LM (which is the one trained for post-FT VT) may differ. We leave this extended analysis for future work.

### 5.2 Classification Tasks: Sentiment & NLI

Figure 5 and Figure 6 show the results of XLM-R on sentiment and NLI. In NLI, we can see that post/pre-FT VT both can reduce the vocabulary to 30K (**39%** of the original XLM-R in the parameter

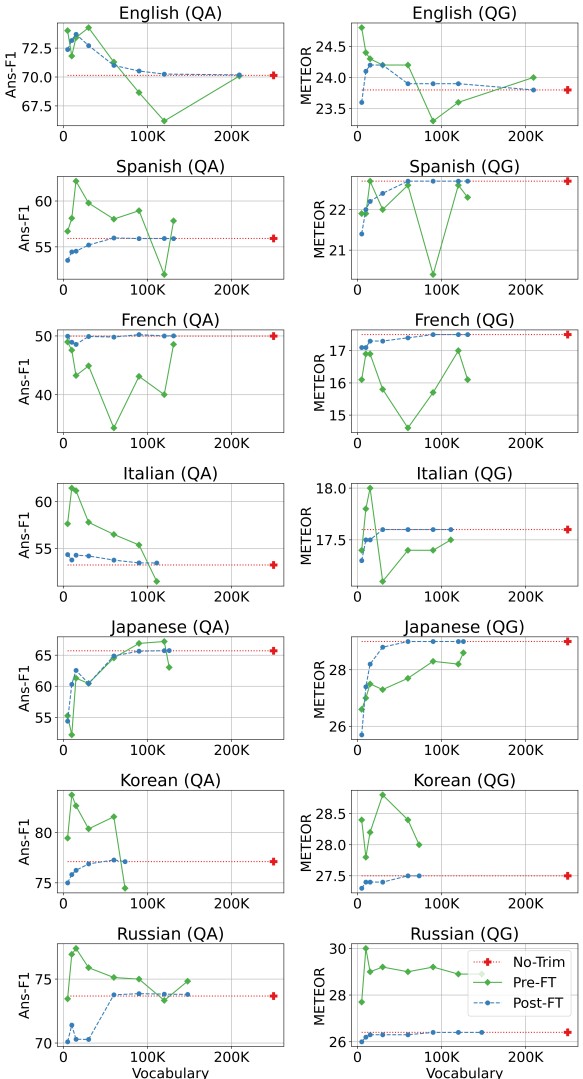

Figure 4: QG (METEOR) and QA (Ans-F1) results for mT5 with pre/post-FT VT for different vocabulary sizes compared to the original multilingual LM (No-Trim).

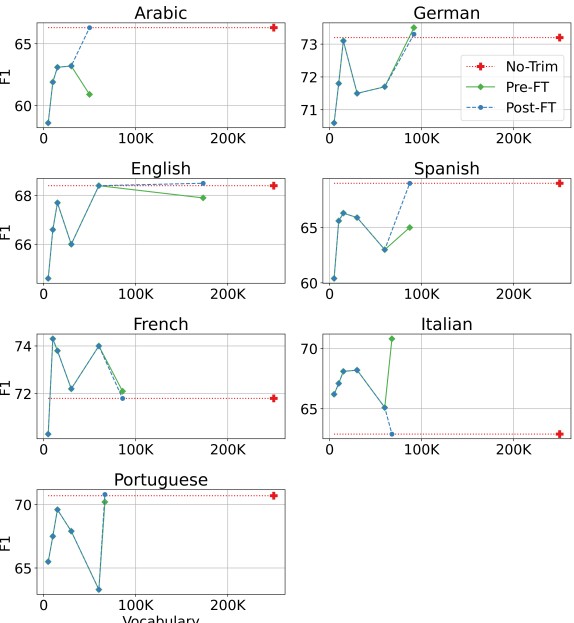

Figure 5: Sentiment analysis macro-F1 results of XLM-R with pre/post-FT VT for different vocabulary sizes compared to No-Trim.

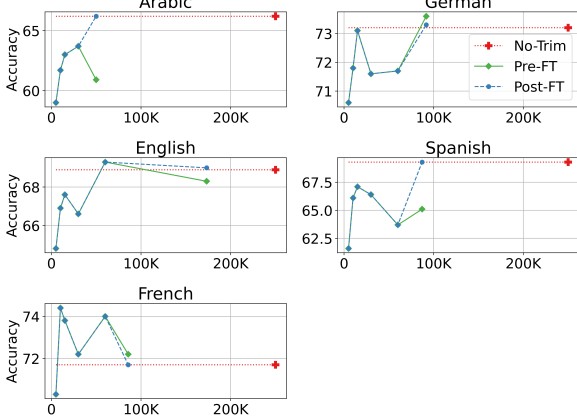

Figure 6: NLI accuracy of XLM-R with pre/post-FT VT for different vocabulary sizes compared to No-Trim.

size) without any decrease except 0.3% of pre-FT VT for German, and there is no decrease more than 0.4% even with top-15K of post-FT VT. In sentiment analysis, pre-FT VT with top-10K (**33%** of the original XLM-R in the parameter size) can retain the accuracy of the No-Trim baseline in French and Italian. Moreover, post-FT VT with 30K can retain the original F1 score without a major drop in sentiment analysis, yet the decrease in F1 score is slightly more prominent than NLI (1.1% in Arabic sentiment analysis).

The sentiment analysis datasets are collected from Twitter, so one dataset in a single language can contain tokens from other languages (hashtags or named-entities, or even code-switching). In contrast, XNLI translates English NLI into other languages, so there is less chance for a dataset to

contain tokens from other languages. This can explain the effectiveness of top-$n$ VT in NLI compared with sentiment analysis, as smaller values of $n$ should result in a vocabulary with fewer tokens from the other languages, which limits the ability of the models to handle foreign tokens.

# 6 Monolingual vs. Multilingual LMs: The Case of Social Bias

There has been extensive literature in NLP comparing monolingual and multilingual LMs (Muller et al., 2021; Goyal et al., 2021). As for the performance, there is no clear consensus on which type is better for certain languages, tasks or settings. How-

ever, there are other important factors that play a role in this comparison. First, monolingual models tend to have a smaller vocabulary size, which makes them more practical. In contrast, a single multilingual model can be used for a large number of languages. Moreover, multilingual LMs are less prone to capture and carry cultural- or language-dependant biases. This is due to the combination of languages and cultures into a single model, which makes it less biased toward specific cultures (Liang et al., 2020; Ahn and Oh, 2021). Prior works have shown that different types of biases consistently appear in language-specific models (Nadeem et al., 2021; Nangia et al., 2020; Blodgett et al., 2021; Dhamala et al., 2021; Kaneko et al., 2022; Zhou et al., 2022). While the comparison of monolingual and multilingual LMs is not the main focus of this paper, this analysis is certainly relevant. Trimming the vocabulary of a multilingual model essentially makes the model smaller, and therefore alleviates one of the main drawbacks of using multilingual language models on language-specific tasks, which is its larger size. On top of that, this strategy enables the usage of monolingual models with potentially less social bias. In the following, we present a comparison of monolingual and multilingual LMs (both trimmed and not trimmed) in terms of social bias and general performance.

## 6.1 Experimental setting

**Social bias datasets.** To study the effect of VT on social bias existing in pre-trained LMs, we first conduct experiments on two commonly used social bias evaluation datasets for masked LMs: StereoSet (SS; Nadeem et al., 2021)[5] and crowdsourced stereotype pairs benchmark (CP; Nangia et al., 2020)[6]. SS consists of associative contexts covering four types of social biases: race, gender, religion, and profession; while CP is crowdsourced and annotated by workers in the United States, which contains nine types of social biases: race, gender, sexual orientation, religion, age, nationality, disability, physical appearance, and socioeconomic status. In order to further investigate the impact of pre/post-FT VT on LMs, we trim and fine-tune models on sentiment analysis with different orders and evaluate the social bias in such models on the Equity Evaluation Corpus (EEC; Kiritchenko and

Mohammad, 2018)[7] considering two bias types: gender and race. The EEC dataset was specifically with the aim to examine social bias for sentiment analysis systems.

**Evaluation metrics.** We compare the pseudo-likelihood scores returned by each model for stereotypical and anti-stereotypical sentences using AULA (All Unmasked Likelihood with Attention weights) (Kaneko and Bollegala, 2022).[8]. AULA has been shown to be robust against the frequency biases of the masked tokens and provides a more reliable assessment in contrast to alternative metrics when evaluating social biases in masked language models (MLMs). Given a sentence pair in the test dataset: "My ***mom*** spent all day cooking for Thanksgiving" vs. "My ***dad*** spent all day cooking for Thanksgiving", the first sentence is considered as stereotypical while the second one is anti-stereotypical. AULA computes the percentage of stereotypical sentences preferred by the MLM over anti-stereotypical ones as the bias score. An MLM is considered to be unfairly biased if it returns higher pseudo-loglikelihood scores for stereotypical sentences than the corresponding anti-stereotypical sentences. The AULA score falls within the range of [0,100] and an unbiased model would return a bias score close to 50. On the other hand, a bias score greater than or less than 50 indicates the bias direction towards the stereotype or anti-stereotype, respectively. Since the original AULA is not fitted to evaluate fine-tuned models, we adapt AULA to the EEC dataset obtain the bias score for the LMs fine-tuned on sentiment analysis, and denote this metric as EEC-AULA. Specifically, given a model that predicts sentiment labels (e.g., positive, neutral, negative) to sentences, we consider the percentage of stereotypical test sentences with a more negative label over anti-stereotypical ones as the corresponding bias evaluation measure.

**General performance.** As a proxy to test the general performance, we use the general language understanding evaluation (GLUE; Wang et al., 2018) benchmark.[9] We acknowledge the limitations of using this benchmark to draw reliable conclusions at large (Ethayarajh and Jurafsky, 2020)

---

[5]https://github.com/moinnadeem/StereoSet
[6]https://github.com/nyu-mll/crows-pairs

[7]https://saifmohammad.com/WebPages/Biases-SA.html
[8]https://github.com/kanekomasahiro/evaluate_bias_in_mlm
[9]https://gluebenchmark.com/; Models are tested on the development sets of each task.

| Base Model | Vocab Trimming | Vocabulary | | Social Bias | | | | General Performance |
| | | Pre-FT | Post-FT | AULA | | EEC-AULA | | GLUE |
| | | | | CP | SS | Gender | Race | |
| Monolingual (RoBERTa) | - | 50K | 50K | 58.1 | 58.8 | 64.8 | 85.7 | 78.0 |
| Multilingual (XLM-R) | - | 250K | 250K | 49.5 | 54.9 | 44.3 | 62.0 | 77.9 |
| | Pre-FT VT (EN) | 173K | 173K | 49.5 | 54.9 | 42.5 | 56.9 | 78.0 |
| | Post-FT VT (EN) | 250K | 173k | 49.5 | 54.9 | 44.3 | 62.0 | 77.9 |
| | Pre-FT VT (50K) | 50K | 50K | 49.3 | 55.0 | 41.0 | 62.4 | 76.9 |
| | Post-FT VT (50K) | 250K | 50K | 49.5 | 54.9 | 44.3 | 62.0 | 77.9 |

Table 3: Results of pre/post-FT VT models compared with the original monolingual and multilingual models on two social bias analysis benchmarks (AULA for pre-trained masked language models and EEC-AULA for models fine-tuned on sentiment analysis) and the GLUE benchmark. The VT models are trimmed on English vocabulary with different vocabulary sizes: EN (full English vocabulary) and 50K (top 50K subword tokens). Note that for post-FT VT, the results on AULA are exactly the same as the original XLM-R. The green and red colours represent the social bias towards anti-stereotypical sentences (scores lower than 50) and stereotypical sentences (scores higher than 50), respectively. The lighter colour indicates less social bias observed in the LM.

but it nevertheless provides a good proxy for understanding the overall performance of comparable models in standard NLP tasks. Moreover, these experiments are only aimed at analysing the effect of vocabulary trimming on general performance.

**Models.** We compute the bias scores of RoBERTa (Liu et al., 2019) as base monolingual LM, and XLM-R (Conneau et al., 2019) as its multilingual counterpart (they have been trained with the same architecture and in an overlapping corpus. We explore two VT settings to be applied to XLM-R: XLM-R with the standard VT including the full English vocabulary (VT XLM-R) and XLM-R with VT for top-50K English vocabulary (top-50K VT XLM-R), which is the same vocabulary size as the monolingual RoBERTa model. Our experiments are based both on masked language models on AULA (in which the post-VT does not have any effect) and models fine-tuned on the sentiment analysis presented in § 4.1 on EEC-AULA, as well as on the corresponding GLUE training sets.

## 6.2 Results

Table 3 shows the performance of pre-FT and post-FT VT models against the original monolingual and multilingual LMs on social bias evaluation datasets and the GLUE benchmark. Both AULA and GLUE results are computed using the LMs without fine-tuning (i.e., RoBERTa, XLM-R, VT XLM-R, and top-50K VT XLM-R), whereas the EEC-AULA results are computed using the models applying VT and fine-tuning strategies. We observe that the monolingual model contains the highest levels of social biases compared to the multilin-

gual models with different settings. In particular, RoBERTa obtains the overall highest bias score on the EEC dataset after fine-tuning, with an alarmingly high 85.7 score on race.[10] On the other hand, compared to the original XLM-R, there is no significant change in performance on social biases and GLUE evaluation tasks for pre-FT VT and post-FT VT models. This is important as we can apply the proposed VT method to any multilingual LM, obtaining a monolingual one with consistent performance on the GLUE benchmark and less social biases than the original monolingual model pre-trained in the target language, without using any ad-hoc debiasing methods.

## 7 Discussion

**Vocabulary trimming before and after fine-tuning.** According to the results, pre-FT VT appears to be generally more effective in classification tasks (see section 4.2.2). For generation tasks, both pre/post-FT VT robustly retain the original performance while being able to considerably reduce the model size (see section 4.2.1). As a guideline to choose the type of VT in such a case, post-FT VT should be more suitable if one already has a fine-tuned model, as no additional training is needed for this case. Moreover, post-FT is more robust as a compression mechanism as the performance is largely maintained with respect to that of the original multilingual LM. On the other hand, if one needs to fine-tune a model from scratch and the computation resources are limited, we recommend

---

[10]Appendix C includes a breakdown by type of the social bias results.

exploring pre-FT VT, as fine-tuning on a trimmed LM should be more efficient due to its smaller vocabulary and parameters and, in some cases, can lead to better overall results. However, this process has to be done carefully as the set of optimal parameters could differ from the original multilingual LM fine-tuning process.

**Monolingual and multilingual LM comparison.** While in this paper we have not compared monolingual and multilingual models, the question would be whether we need vocabulary trimming strategies in a world where monolingual LMs exist. In this case, a monolingual model may perform similarly to a multilingual model (Goyal et al., 2021). However, the multilingual model is often larger mainly due to larger vocabulary storage requirements. In contrast, our proposed VT technique does not require any extra LM training or computational resources. Indeed, only a multilingual LM is needed and we can induce multiple smaller language-specific monolingual models. This may reduce the carbon footprint overall and especially help with less-resource languages when a high-quality monolingual model does not exist. Finally, our social bias analysis see § 6 shows how monolingual models exhibit larger social biases (especially racial) than a VT-induced multilingual LM. This is consistent with prior work suggesting that a multilingual LM has been trained with more languages, and hence more cultural variety, and these diverging viewpoints can compensate each other (Ahn and Oh, 2021).

## 8    Conclusion

In this paper, we proposed *vocabulary-trimming* (VT), a method to reduce the vocabulary of a multilingual LM to a vocabulary specific to any given target language. VT can induce a monolingual LM in the target language by leveraging an existing multilingual LM. The main advantage of this filtering step is the reduced size, as well as avoiding having to train monolingual LMs from scratch, which would be computationally demanding. Our experiments show how VT can retain the high performance of the original multilingual LM, while largely reducing the model size. For all languages evaluated, a 35% compression rate proves sufficient to keep the original performance of the larger mT5 multilingual LM in both QA and QG, with a similar 39% in NLI and 55% in sentiment analysis with XLM-R. Interestingly, in some cases, the

compressed LM can even achieve better results than the original larger model when trimmed before fine-tuning. Since the main goal of the paper was to compress a multilingual LM while keeping its original performance, we leave the analysis of this behaviour for future work.

## Limitations

We have not tested our methodology in truly low-resource languages. Because of this, there could be a different behaviour when we apply VT to a language with lower resources or that is poorly represented in the underlying training corpus. The LMs we used in the paper limited their size up to 600M, and we have not considered larger models such as mT5$_{XXL}$ or BLOOM (Scao et al., 2022), due to our limited computational resources. As the language-specific corpus to compute frequency, we employ mC4, which is one of the largest multilingual corpora. Nonetheless, this is used as a proxy and having access to the full multilingual model could give potentially better results.

Similarly, we acknowledge the limitations of the analysis comparing multilingual and monolingual models in terms of social bias. Due to the evaluation data available and the existence of comparable monolingual and multilingual LMs, the evaluation is focused on English only and the results could differ for other languages. Moreover, there are other types of biases not covered in this evaluation.

## Ethics Statement

Pre-trained LMs are known to contain undesirable biases to generate toxic contents in some edge cases (Schick et al., 2021), so the resulting models could inherit such biases. While we have not analysed in detail the output of all models in the tasks evaluated, in this paper we have made an attempt to study this effect in terms of social biases for both base pre-trained LMs and fine-tuned LMs.

## Acknowledgements

Jose Camacho-Collados and Yi Zhou are supported by a UKRI Future Leaders Fellowship.

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

| Vocab. | No-Trim (250K) | 5K | 10K | 15K | 30K | 60K |
|--------|---------------|------|------|------|------|------|
| Param. | 278M | 89M | 93M | 97M | 109M | 132M |
| **Post-FT (Sentiment)** | | | | | | |
| AR | **66.3** | 64.5 | 64.5 | 65.9 | 65.9 | - |
| DE | 73.2 | 70.4 | 72.4 | 72.1 | **73.7** | 73.3 |
| EN | 68.4 | 64.0 | 66.5 | 67.4 | **68.6** | 68.5 |
| ES | **69.0** | 66.2 | 67.2 | 67.8 | 68.4 | **69.0** |
| FR | **71.8** | 68.6 | 71.4 | 71.7 | 71.6 | **71.8** |
| IT | 62.9 | 60.8 | 61.9 | **63.5** | 62.3 | 62.9 |
| PT | 70.7 | 63.6 | 65.9 | 68.2 | 69.4 | **70.8** |
| **Pre-FT (Sentiment)** | | | | | | |
| AR | **66.3** | 58.6 | 61.9 | 63.1 | 63.2 | - |
| DE | **73.2** | 70.6 | 71.8 | 73.1 | 71.5 | 71.7 |
| EN | **68.4** | 64.6 | 66.6 | 67.7 | 66.0 | **68.4** |
| ES | **69.0** | 60.4 | 65.6 | 66.3 | 65.9 | 63.0 |
| FR | 71.8 | 70.3 | **74.3** | 73.8 | 72.2 | 74.0 |
| IT | 62.9 | 66.2 | 67.1 | 68.1 | **68.2** | 65.1 |
| PT | **70.7** | 65.5 | 67.5 | 69.6 | 67.9 | 63.3 |
| **Post-FT (NLI)** | | | | | | |
| AR | 75.7 | 75.0 | **75.8** | **75.8** | 75.7 | - |
| DE | **79.9** | 76.6 | 78.9 | 79.6 | **79.9** | **79.9** |
| EN | **84.6** | 80.5 | 83.6 | 84.3 | **84.6** | **84.6** |
| ES | **79.8** | 75.3 | 78.4 | 79.4 | **79.8** | **79.8** |
| FR | 80.1 | 77.2 | 80.0 | 80.1 | **80.2** | 80.1 |
| **Pre-FT (NLI)** | | | | | | |
| AR | **75.7** | 73.0 | 75.0 | 75.4 | **75.7** | - |
| DE | **79.9** | 77.4 | 79.0 | 79.0 | 79.6 | 79.7 |
| EN | 84.6 | 83.3 | 84.7 | 84.4 | **85.1** | 84.3 |
| ES | 79.8 | 78.9 | 79.1 | 79.2 | **81.0** | 79.7 |
| FR | **80.1** | 77.3 | 78.9 | 78.2 | **80.1** | 78.7 |

Table 4: Results of XLM-R for sentiment analysis (macro F1) and NLI (accuracy) with different top-$n$ vocabulary size at VT, where the best results in each LM and language are in the bold characters.

*(Volume 1: Long Papers)*, pages 1924–1935, Dublin, Ireland. Association for Computational Linguistics.

## A Top-$n$ VT of XLM-R

Table 4 shows the results of XLM-R fine-tuned on sentiment and NLI with post/pre-VT for different top-$n$.

## B Top-$n$ VT of mT5

Table 5 shows the results of mT5 fine-tuned on QA and QG with post/pre-VT for different top-$n$.

## C Details of Results on Social Bias Evaluation

Table 6 shows the details of social bias evaluation (EEC dataset) regarding each emotion type observed in the LMs fine-tuned on sentiment analysis. Table 7 shows the details of social bias regarding each bias type in both CP and SS datasets observed in the comparison LMs.

| Vocab. Param. | No-Trim (250K) 300M | 5K 49M | 10K 54M | 15K 59M | 30K 74M | 60K 105M | 90K 136M | 120K 166M |
|---|---|---|---|---|---|---|---|---|
| **Post-FT (QA)** | | | | | | | | |
| EN | 70.1 / 55.5 | 72.4 / 59.7 | 73.2 / 60.5 | **73.7 / 60.9** | 72.7 / 58.8 | 71.0 / 56.4 | 70.5 / 55.8 | 70.3 / 55.6 |
| ES | 55.9 / 34.7 | 53.5 / 32.5 | 54.4 / 33.1 | 54.5 / 33.2 | 55.2 / 33.8 | **56.0 / 34.8** | 55.9 / **34.8** | 55.9 / 34.7 |
| FR | 50.0 / 30.9 | 50.0 / **31.5** | 48.9 / 29.9 | 48.6 / 29.3 | 49.9 / 30.1 | 49.8 / 30.5 | **50.2** / 30.9 | 50.0 / 30.8 |
| IT | 53.2 / 37.6 | **54.3 / 39.6** | 53.8 / 38.4 | 54.3 / 38.8 | 54.2 / 38.5 | 53.7 / 38.1 | 53.4 / 37.8 | - |
| JA | **65.7 / 65.7** | 54.4 / 54.4 | 60.3 / 60.3 | 62.6 / 62.6 | 60.4 / 60.4 | 64.9 / 64.9 | 65.6 / 65.6 | **65.7 / 65.7** |
| KO | 77.1 / 70.6 | 75.0 / 67.8 | 75.8 / 68.7 | 76.2 / 69.3 | 76.9 / 70.2 | **77.3 / 70.7** | - | - |
| RU | 73.7 / **51.4** | 70.1 / 48.8 | 71.4 / 49.5 | 70.3 / 47.6 | 70.3 / 47.9 | 73.8 / 51.1 | **73.9 / 51.4** | 73.8 / **51.4** |
| **Post-FT (QG)** | | | | | | | | |
| EN | 23.8 / **90.0** | 23.6 / 89.8 | 24.1 / 89.9 | **24.2 / 90.0** | 24.2 / 90.0 | 23.9 / 90.0 | 23.9 / 90.0 | 23.9 / 90.0 |
| ES | **22.7 / 84.1** | 21.4 / 83.6 | 22.0 / 83.8 | 22.2 / 83.9 | 22.4 / 84.0 | **22.7 / 84.1** | **22.7 / 84.1** | **22.7 / 84.1** |
| FR | **17.5 / 80.7** | 17.1 / 80.3 | 17.1 / 80.4 | 17.3 / 80.5 | 17.3 / 80.5 | 17.4 / 80.6 | **17.5 / 80.7** | **17.5 / 80.7** |
| IT | **17.6 / 80.8** | 17.3 / 80.7 | 17.5 / **80.8** | 17.5 / **80.8** | **17.6** / 80.7 | **17.6 / 80.8** | **17.6 / 80.8** | - |
| JA | **29.0 / 80.9** | 25.7 / 79.2 | 27.4 / 80.1 | 28.2 / 80.5 | 28.8 / **80.9** | **29.0 / 80.9** | **29.0 / 80.9** | **29.0 / 80.9** |
| KO | **27.5 / 82.9** | 27.3 / 82.8 | 27.4 / 82.9 | 27.4 / 82.9 | 27.4 / 82.9 | **27.5 / 83.0** | - | - |
| RU | **26.4 / 84.3** | 26.0 / 84.0 | 26.2 / 84.2 | 26.3 / 84.2 | 26.3 / 84.2 | 26.3 / **84.3** | **26.4 / 84.3** | **26.4 / 84.3** |
| **Pre-FT (QA)** | | | | | | | | |
| EN | 70.1 / 55.5 | 74.0 / **61.3** | 71.8 / 59.1 | 73.4 / 60.7 | **74.3 / 61.3** | 71.3 / 56.5 | 68.6 / 54.3 | 66.2 / 52.1 |
| ES | 55.9 / 34.7 | 56.7 / 36.7 | 58.1 / 37.5 | **62.2 / 41.4** | 59.8 / 40.1 | 58.0 / 37.0 | 58.9 / 38.1 | 52.0 / 32.7 |
| FR | **50.0 / 30.9** | 49.0 / **32.7** | 47.6 / 30.2 | 43.3 / 26.9 | 44.9 / 27.4 | 34.3 / 20.0 | 43.1 / 24.1 | 40.0 / 23.7 |
| IT | 53.2 / 37.6 | 57.6 / 43.7 | **61.5 / 46.8** | 61.2 / 45.6 | 57.8 / 42.1 | 56.5 / 40.5 | 55.4 / 39.2 | - |
| JA | 65.7 / 65.7 | 55.3 / 55.3 | 52.2 / 52.2 | 61.3 / 61.3 | 60.4 / 60.4 | 64.6 / 64.6 | 66.9 / 66.9 | **67.2 / 67.2** |
| KO | 77.1 / 70.6 | 79.5 / 72.6 | **83.7 / 77.7** | 82.7 / 76.4 | 80.4 / 73.7 | 81.6 / 75.1 | - | - |
| RU | 73.7 / 51.4 | 73.5 / 51.7 | 76.9 / 56.4 | **77.4 / 56.9** | 75.9 / 54.2 | 75.1 / 53.0 | 75.0 / 53.0 | 73.3 / 51.4 |
| **Pre-FT (QG)** | | | | | | | | |
| EN | 23.8 / **90.0** | **24.8 / 90.0** | 24.4 / 89.9 | 24.3 / 89.9 | 24.2 / **90.0** | 24.2 / **90.0** | 23.3 / 89.9 | 23.6 / 89.8 |
| ES | **22.7 / 84.1** | 21.9 / 84.1 | 21.9 / 83.9 | **22.7** / 83.7 | 22.0 / 84.3 | 22.6 / **84.4** | 20.4 / 79.6 | 22.6 / 84.1 |
| FR | **17.5 / 80.7** | 16.1 / 79.0 | 16.9 / 79.9 | 16.9 / 79.5 | 15.8 / 78.8 | 14.6 / 78.0 | 15.7 / 79.0 | 17.0 / 79.2 |
| IT | 17.6 / 80.8 | 17.4 / 80.4 | 17.8 / **81.1** | **18.0** / 80.8 | 17.1 / 80.6 | 17.4 / 80.9 | 17.4 / 80.8 | - |
| JA | **29.0 / 80.9** | 26.6 / 79.8 | 27.0 / 79.2 | 27.5 / 79.7 | 27.3 / 79.9 | 27.7 / 80.3 | 28.3 / 80.8 | 28.2 / 80.2 |
| KO | 27.5 / 82.9 | 28.4 / 83.4 | 27.8 / 83.4 | 28.2 / **84.1** | **28.8** / 83.4 | 28.4 / 83.4 | - | - |
| RU | 26.4 / 84.3 | 27.7 / 85.9 | **30.0 / 86.6** | 29.0 / 86.5 | 29.2 / 86.3 | 29.0 / 86.6 | 29.2 / **86.7** | 28.9 / 86.0 |

Table 5: Results of mT5 for QA (Ans-F1/EM) and QG (MTR/BS) with different top-$n$ vocabulary size at VT, where the best results in each LM and language are in the bold characters.

| | Model | AULA-EEC | Anger | Sadness | Fear | Joy | No Emotion Type |
|---|---|---|---|---|---|---|---|
| **Gender Bias** | FT RoBERTa | 64.8 | 50.0 | 67.8 | 71.5 | 64.3 | 60.8 |
| | FT XLM-R | 44.3 | 46.9 | 39.4 | 51.4 | 34.4 | 39.7 |
| | Pre-FT XLM-R (EN) | 42.5 | 33.8 | 55.8 | 45.1 | 40.7 | 42.7 |
| | Post-FT XLM-R (EN) | 44.3 | 46.9 | 39.4 | 51.4 | 34.4 | 39.7 |
| | Pre-FT XLM-R (50k) | 44.3 | 46.9 | 39.4 | 51.4 | 34.4 | 39.7 |
| | Post-FT XLM-R (50k) | 41.0 | 41.2 | 45.8 | 47.2 | 36.9 | 37.2 |
| **Race Bias** | FT RoBERTa | 85.7 | 50.0 | 88.0 | 65.4 | 50.0 | 100.0 |
| | FT XLM-R | 62.0 | 74.5 | 32.1 | 47.3 | 74.6 | 17.4 |
| | Pre-FT VT XLM-R (EN) | 56.9 | 33.7 | 58.8 | 69.2 | 60.1 | 50.0 |
| | Pre-FT VT XLM-R (EN) | 62.0 | 74.5 | 32.1 | 47.3 | 74.6 | 17.4 |
| | Pre-FT VT XLM-R (50k) | 62.0 | 74.5 | 32.1 | 47.3 | 74.6 | 17.4 |
| | Post-FT VT XLM-R (50k) | 62.4 | 55.7 | 70.3 | 70.9 | 59.1 | 59.0 |

Table 6: Social bias scores of LMs fine-tuned on sentiment analysis on each emotion type w/wo VT on the EEC dataset.

|  |  | RoBERTa | XLM-R | VT XLM-R (EN) | VT XLM-R (50K) |
|---|---|---|---|---|---|
| CP Dataset | AULA | 58.1 | 49.5 | 49.5 | 49.3 |
|  | Age | 59.8 | 49.4 | 49.4 | 49.4 |
|  | Disability | 68.3 | 66.7 | 66.7 | 68.3 |
|  | Gender | 53.4 | 48.5 | 48.5 | 48.1 |
|  | Nationality | 56.6 | 44.0 | 44.0 | 44.7 |
|  | Physical-Appearance | 52.4 | 58.7 | 58.7 | 55.6 |
|  | Race-Color | 56.8 | 43.8 | 43.8 | 44.0 |
|  | Religion | 53.3 | 54.3 | 54.3 | 53.3 |
|  | Sexual-Orientation | 67.9 | 54.8 | 54.8 | 54.8 |
|  | Socioeconomic | 66.3 | 58.1 | 58.1 | 57.6 |
| SS Dataset | AULA | 58.8 | 54.9 | 54.9 | 55.0 |
|  | Gender | 62.4 | 55.3 | 55.3 | 54.9 |
|  | Profession | 60.4 | 54.3 | 54.3 | 54.6 |
|  | Race | 57.0 | 56.0 | 56.0 | 56.0 |
|  | Religion | 54.4 | 46.8 | 46.8 | 46.8 |

Table 7: Social bias scores of VT LMs compared to their original ones on both CP and SS datasets for each bias type.