# OpenReview forum: "Efficient Multilingual Language Model Compression through Vocabulary Trimming"
_EMNLP/2023/Conference — EMNLP 2023 Findings_

### Official Review · Reviewer_dgGG · 2023-08-01

**Soundness:** 4

**Excitement:**

3: Ambivalent: It has merits (e.g., it reports state-of-the-art results, the idea is nice), but there are key weaknesses (e.g., it describes incremental work), and it can significantly benefit from another round of revision. However, I won't object to accepting it if my co-reviewers champion it.

**Paper Topic And Main Contributions:**

This paper looks at the problem of compressing the model size of a multilingual language model when its intended downstream use is on a monolingual task. The method the authors present is called vocabulary trimming (VT), which involves deleting tokens from the vocabulary which do not feature in the target language.

The main contributions of this paper are the following:

- Explanation of VT, a method for removing tokens from the multilingual vocabulary which are not relevant to the target language.
- Extensive empirical work showing that VT has a minimal impact on downstream monolingual tasks (question answering, question generation, sentiment analysis, natural language inference)
- Additional empirical work exploring the impact of vocabulary size and when VT is carried out
- Exploration of VT as a debiasing technique

**Reasons To Accept:**

1) The paper presents extensive empirical work backing up their claims covering multiple high-resource languages.
2) Their figures are particularly clear and helpful for explaining their method
3) Their method is well-situated in the current literature and they cover a wide range of related work

**Reasons To Reject:**

1) From reading the introduction and section 4, I was unclear why VT was an improvement over simply using an existing monolingual model. It would be good to have a better justification/explanation in the introduction and/or a comparison with monolingual models in the analysis in section 4
2) On lines 103-4, the authors say it is a "natural question" to ask if VT impacts bias levels. As someone unfamiliar with this field, I found the switch to VT's impact on bias jarring. It would help if the authors could make it clearer how they made this jump e.g. comparison with existing literature

**Reproducibility:**

4: Could mostly reproduce the results, but there may be some variation because of sample variance or minor variations in their interpretation of the protocol or method.

**Reviewer Confidence:**

2: Willing to defend my evaluation, but it is fairly likely that I missed some details, didn't understand some central points, or can't be sure about the novelty of the work.

**Typos Grammar Style And Presentation Improvements:**

- line 1: missing bracket after "(LMs)"
- Figure 4: graphs are too small to interpret easily. Perhaps these results would be easier to understand in a table?
- Table 1: having two metrics side by side was confusing and overloaded the table. Given they both say roughly the same thing, perhaps just pick one and put the full results for the other in the appendix
- Tables 1, 2, 3: There is too much information here to parse easily. Given the main point here is that there is not much change over the baseline, a delta might be easier to understand than a raw result.

---

> ### Author Rebuttal · Authors · 2023-08-28
>
> [Reason 1] From reading the introduction and section 4, I was unclear why VT was an improvement over simply using an existing monolingual model. It would be good to have a better justification/explanation in the introduction and/or a comparison with monolingual models in the analysis in section 4
> [Response] As we mentioned in the introduction, the reasons why we need VT are two-fold. First, multilingual LMs are shown to be competitive to its monolingual counterpart (XLM-R, XLM-V, mT5). Second, we do not have mono-lingual LMs in many languages, especially when it comes to encoder-decoder LMs (e.g. T5). Accordingly, we need to fine-tune multilingual LMs in many cases, but they are often larger than its monolingual counterpart due to the multilingual embedding matrix. Here, we propose VT to remove unused vocabulary to make the multilingual LM as small as the monolingual counterpart, when you fine-tune it in a specific language. We unfortunately do not have a result more than English in section 4.
>
>
> [Reason 2] On lines 103-4, the authors say it is a "natural question" to ask if VT impacts bias levels. As someone unfamiliar with this field, I found the switch to VT's impact on bias jarring. It would help if the authors could make it clearer how they made this jump e.g. comparison with existing literature
> [Response] As we mentioned in line 97-103, prior works have shown that pretrained language models contain worrying level of social bias (May et al., 2019; Kurita et al., 2019; Kaneko and Bollegala, 2021). In addition, Kaneko et al. (2022) showed that the level of social bias in masked language models increases during fine-tuning. For this reason, we investigate if VT has impacts on social bias in language models.

---

### Official Review · Reviewer_9JRE · 2023-08-04

**Soundness:** 4

**Excitement:**

3: Ambivalent: It has merits (e.g., it reports state-of-the-art results, the idea is nice), but there are key weaknesses (e.g., it describes incremental work), and it can significantly benefit from another round of revision. However, I won't object to accepting it if my co-reviewers champion it.

**Paper Topic And Main Contributions:**

This paper introduces vocabulary trimming (VT) which aims to reduce a multilingual LM to a monolingual LM. The method involves trimming the vocabulary of a multilingual LM, using a target language's monolingual corpus, to include only tokens relevant to the target language. They consider applying VT before or after finetuning, and show the tradeoffs between both methods. Furthermore, they compare a trimmed multilingual model to a monolingual model trained from scratch and show that the trimmed model has less bias.

**Questions For The Authors:**

- Why report only accuracy for NLI?
- Please report effect of VT on low resourced languages.

**Reasons To Accept:**

- Fairly efficient method because you can potentially extract several monolingual LMs from a single multilingual LM.
-

**Reasons To Reject:**

- No analysis of tradeoff between extracting monolingual LM and potentially losing helpful cross-lingual transfer.
- Given that it is a long paper, it is missing important analysis such as effect of monolingual corpora domain and size, result on resourced languages and effect of % of target language composition in original multilingual corpora.
- Only evaluate on English
- Did not evaluate on larger LMs.

**Reproducibility:**

4: Could mostly reproduce the results, but there may be some variation because of sample variance or minor variations in their interpretation of the protocol or method.

**Reviewer Confidence:**

4: Quite sure. I tried to check the important points carefully. It's unlikely, though conceivable, that I missed something that should affect my ratings.

---

> ### Author Rebuttal · Authors · 2023-08-28
>
> [Reason 1] No analysis of tradeoff between extracting monolingual LM and potentially losing helpful cross-lingual transfer.
> [Response] We focus on adapting multilingual LMs to a specific language, in other word, localising multilingual LM to a single target language, to perform well on that language, while reducing the model size largely. We focus on monolingual tasks, rather than multilingual or cross-lingual application. Certainly, the trimmed LMs are no longer useful for cross-lingual/multilingual application, since we drop the vocabularies not used in the target language, but that is not what we care most about in our paper.
>
>
> [Reason 2] Given that it is a long paper, it is missing important analysis such as effect of monolingual corpora domain and size, result on resourced languages and effect of % of target language composition in original multilingual corpora.
> [Response] We focus on relatively high-resource languages in our paper, where the portions of those languages in the pre-training corpus of multilingual LMs are large. Within such high-resource languages, we assume that the difference among the languages in terms of the size for the pre-training corpus, has limited effect on the performance. In fact, our method works well among all the languages we considered in the paper. Perhaps, in a low-resource language, the performance could correlate with the size of that language-specific texts in the pre-training corpus of the multilingual LMs, yet it is out of our scope as mentioned in the limitations section.
>
>
> [Reason 3] Only evaluate on English
> [Response] Our study deals with four multilingual language models (mT5/mBART/XLM-R/XLM-V), and we have 7 languages (en/es/fr/ko/de/ja/ru) for QA/QG task, as well as 6 languages (ar/de/en/es/it/pt) for NLI/sentiment analysis. We would appreciate if the reviewer could elaborate the reason in detail.
>
>
> [Reason 4] Did not evaluate on larger LMs.
> [Response] As we mentioned explicitly in our limitations section, the application of our method on the larger LM such as BLOOM and mT5-XXL is out of the scope for our paper.
>
>
> [Q1]: Why report only accuracy for NLI?
> [Answer] We rely on accuracy, because it is the metric used in the original paper. We do not study NLI but we consider NLI as one of the applications, so it should not be a problem to report a standard metric used in the task setting.
>
>
> [Q2]: Please report effect of VT on low resourced languages.
> [Answer] Thank you for the suggestion. As we mentioned explicitly in our limitations section, the application of our method on the low-resource language is out of the scope for our paper.

---

### Official Review · Reviewer_XE1g · 2023-08-05

**Soundness:** 4

**Excitement:**

3: Ambivalent: It has merits (e.g., it reports state-of-the-art results, the idea is nice), but there are key weaknesses (e.g., it describes incremental work), and it can significantly benefit from another round of revision. However, I won't object to accepting it if my co-reviewers champion it.

**Paper Topic And Main Contributions:**

Authors propose a simple vocabulary trimming technique that removes tokens from the vocabulary of multilingual LMs by identifying language-specific tokens and retaining those. Given a large enough text corpus (on which to perform token retention), theoretically there should be no drop in model performance.

**Questions For The Authors:**

- It's unclear why pre-FT VT reduces finetuning time -- if a language can be modeled with a subset of vocabulary V, V_subset, then finetuning would theoretically have the same computational complexity since we only use a subset of the vocabulary anyways (assuming V_subset contains all the tokens we need for that particular language). Is it just the extra time to perform the embedding lookup?
- Is there a performane drop for pre-FT VT, since you can't leverage translate-train-all or cross-lingual transfer from high resource languages?
- Have the authors tried this method for ultra-low-resource languages and on datasets like MasakhaNER?
- What is the size of the per-language text corpora used to perform vocabulary retention?
- Do the authors see a marked increase in OOV after applying this technique?

**Reasons To Accept:**

- This method can significantly reduce the memory footprint of multilingual models, especially smaller multilingual models where a large portion of the parameters are tied to the vocabulary embedding lookup table.
- While the capabilities are of this method are mostly tied to memory reduction

**Reasons To Reject:**

- This method essentially removes the capability of the model to do well in languages other than the intended language. While theoretically sound and would work well on academic datasets, real world text is rife with code-switching and text in multiple languages. This could limit the capabilities of this method.
- Much of the benefits of this method could be potentially covered by just loading the vocabulary embeddings into CPU memory and pushing the required vocabulary embeddings per batch into GPU memory on-the-fly. Have the authors compared the inference latency increases using this method?

**Reproducibility:**

5: Could easily reproduce the results.

**Reviewer Confidence:**

5: Positive that my evaluation is correct. I read the paper very carefully and I am very familiar with related work.

---

> ### Author Rebuttal · Authors · 2023-08-28
>
> We found the reviews highly useful, and really appreciate the effort you put in writing those comments!
>
> [Reason 1] This method essentially removes the capability of the model to do well in languages other than the intended language. While theoretically sound and would work well on academic datasets, real world text is rife with code-switching and text in multiple languages. This could limit the capabilities of this method.
> [Response] Thank you for the useful feedback. We have not tested our method on a dataset with code-switching, so we will add the note to the limitation section. Indeed, to apply vocabulary-trimming on such a corpus consisting of texts from different languages, is a decent direction for future work.
>
>
> [Reason 2] Much of the benefits of this method could be potentially covered by just loading the vocabulary embeddings into CPU memory and pushing the required vocabulary embeddings per batch into GPU memory on-the-fly. Have the authors compared the inference latency increases using this method?
> [Response] That approach should save the GPU memory at the fine-tuning, but the model is still the same size as the original model. Our approach is not only aiming at increasing the training efficiency but also reducing the cost of hosting the model afterward. Since the vocabulary trimming can reduce the model size largely, hosting the model requires less storage and memory to the server. The CPU core can be less powerful too, as the inference takes less time, mainly because the complexity of the lookup over the vocabulary (embedding to token/token to embedding) is reduced by the smaller size of the vocabulary.
>
>
> [Q1] It's unclear why pre-FT VT reduces finetuning time -- if a language can be modeled with a subset of vocabulary V, V_subset, then finetuning would theoretically have the same computational complexity since we only use a subset of the vocabulary anyways (assuming V_subset contains all the tokens we need for that particular language). Is it just the extra time to perform the embedding lookup?
> [Response] Regarding the fine-tuning, the main benefit is from the reduced time in the embedding lookup, as mentioned.
>
>
> [Q2] Is there a performane drop for pre-FT VT, since you can't leverage translate-train-all or cross-lingual transfer from high resource languages?
> [Response] In general, VT can achieve competitive results as the original model, but related to our response to the [Reason 1], the tasks we considered in our paper are all mono-lingual with less code-switching. If we applied the method to the task with mixed languages, VT would limit the performance due to the lack of cross-lingual understanding caused by VT.
>
>
> [Q3] Have the authors tried this method for ultra-low-resource languages and on datasets like MasakhaNER?
> [Response] As we mention in the limitations section, our study is based on high-resource languages at the moment.
>
>
> [Q4] What is the size of the per-language text corpora used to perform vocabulary retention?
> [Response] We rely on the validation split of the mC4 https://aclanthology.org/2021.naacl-main.41/, the training dataset of mT5. It contains 3,083K (en), 416k (es), 331k (fr), 15.8k (ko), 87.4k (ja), 53k (ar), 186k (it), 399k (de), and 169k (pt).
>
>
> [Q5] Do the authors see a marked increase in OOV after applying this technique?
> [Answer] Within the experiments we conducted in the paper, OOV errors are less prominent. In fact, the tokenizer rolls back unknown vocabularies into smaller chunks, so it should not be an issue as long as the text is in the target language. English words, as an example, would split into a sequence of alphabet at the finest, which are often part of the vocabulary after the trimming. Moreover, we trim the vocabulary based on the frequency of them in the language-specific corpus, so in theory, this already reduces the risk of removing important vocabularies, and avoid raising OOV in practice.

---

### Meta-Review · Area_Chair_sZz2 · 2023-09-19

**Recommendation:** 3

**Metareview:**

The reviewers found the paper to be sound, and appreciated the experimental results, such as the result that the vocabulary-trimmed multilingual model shows less bias than a monolingual model trained from scratch.

However, the reviewers ultimately questioned the value of the contribution since 1) it may excessively harm the model in real-world settings, where data is not always cleanly in a single language, and 2) the memory footprint could also be reduced simply by avoiding naively loading the entire embedding table into memory.

---

### Decision · Program_Chairs · 2023-10-07

**Decision:**

Accept-Findings

**Comment:**

The reviewers found the paper to be sound, and appreciated the experimental results, such as the result that the vocabulary-trimmed multilingual model shows less bias than a monolingual model trained from scratch.

However, the reviewers ultimately questioned the value of the contribution since 1) it may excessively harm the model in real-world settings, where data is not always cleanly in a single language, and 2) the memory footprint could also be reduced simply by avoiding naively loading the entire embedding table into memory.